# Novel Pilot-Scale Photocatalytic Nanofiltration Reactor for Agricultural Wastewater Treatment

**DOI:** 10.3390/membranes13020202

**Published:** 2023-02-06

**Authors:** George V. Theodorakopoulos, Michalis K. Arfanis, José Antonio Sánchez Pérez, Ana Agüera, Flor Ximena Cadena Aponte, Emilia Markellou, George Em. Romanos, Polycarpos Falaras

**Affiliations:** 1Institute of Nanoscience and Nanotechnology, National Center of Scientific Research “Demokritos”, Agia Paraskevi, 15310 Athens, Greece; 2Inorganic and Analytical Chemistry Laboratory, School of Chemical Engineering, National Technical University of Athens, Zografou Campus, 9 Iroon Polytechneiou Str., Zografou, 15772 Athens, Greece; 3Solar Energy Research Centre (CIESOL), Joint Centre University of Almería-CIEMAT, Carretera de Sacramento s/n, E-04120 Almería, Spain; 4Chemical Engineering Department, University of Almería, Carretera de Sacramento s/n, E-04120 Almería, Spain; 5Laboratory of Mycology, Scientific Directorate of Phytopathology, Benaki Phytopathological Institute, 8 St. Delta Str., Kifissia, 14561 Athens, Greece

**Keywords:** photocatalytic nanofiltration reactor (PNFR), titanium dioxide, wash-coating technique, nanofiltration, pesticides, wastewater, fruit industry

## Abstract

Nowadays, the increased agro-industrial activities and the inability of traditional wastewater treatment plants (WWTPs) to eliminate recalcitrant organic contaminants are raising a potential worldwide risk for the environment. Among the various advanced water treatment technologies that are lately proposed for addressing this challenge, the development and optimization of an innovative hybrid photocatalytic nanofiltration reactor (PNFR) prototype emerges as a prominent solution that achieves synergistic beneficial effects between the photocatalytic degradation activity and size exclusion capacity for micropollutant molecules. Both these features can be contemporarily endued to a multitude of membrane monoliths. The physicochemical and the photoinduced decontamination properties of the titania materials were firstly determined in the powder form, and subsequently, the structural and morphological characterization of the obtained titania-modified membrane monoliths were accomplished. The PNFR unit can operate at high water recovery and low pressures, exhibiting promising removal efficiencies against Acetamiprid (ACT) and Thiabendazole (TBZ) pesticides and achieving the recycling of 15 m^3^/day of real agro-wastewater. The obtained results are very encouraging, demonstrating the integration of titania photocatalysts in a photocatalytic membrane reactor as a feasible technological solution for the purification of agricultural wastewater.

## 1. Introduction

In recent years, the agro-food industry achieved significant improvement on agricultural production and quality, through the development and the utilization of pesticides. Nevertheless, most of the pesticides are recalcitrant and toxic organic substances, which have the potential to enter the environment even through the effluents of conventional WWTPs, and are accumulated on freshwater resources and soil. The pesticides’ toxicity can cause an adverse impact on nature and the food chain, leading to harmful effects on human health; therefore, intensive actions are required to eliminate the potential risks. Until now, conventional wastewater treatment methods are inefficient to confront pesticides, so new strategies must be developed, evaluated, and materialized in the form of novel water treatment technologies that can be retrofitted into the existing infrastructure [1]. Most of the proposed technologies are very promising, though some drawbacks and weaknesses negate their effectiveness and limit their usage [2]. For example, the sorption with activated carbon generates extra solid waste, which is costly to manage; the membrane filtration (micro-, ultra- and nano-) suffering from fouling needs the periodic regeneration of the membrane; ozonation needs cost-effective equipment; and photo-Fenton and slurry photocatalysis have to separate the catalyst from the treated water downstream of the process. Interestingly, radical improvements could be brought about in wastewater treatment, given that the aforementioned technologies could be appropriately integrated to interact synergistically towards addressing the undesired effects.

In this context, a patented lab-scale reactor was designed for wastewater treatment [3], integrating the most effective water treatment technologies against micropollutants in one reactor module: photocatalysis and filtration. The performance of this hybrid photocatalytic nanofiltration reactor (PNFR) was based on (i) the capacity to irradiate a multitude of photocatalytic surfaces inside the photocatalytic-membrane reactor module contemporarily with the implementation of the tangential flow-filtration process; (ii) the oxidative micropollutant degradation by titania (TiO_2_) photocatalysts and the photoinduced radicalary mechanism during the illumination with light of appropriate wavelength [4]; and (iii) the simultaneous retention of micropollutants and metabolites onto the nanoporous membranes [5]. The novel design of the PNFR permits the coating of the membrane with any type of nanostructured, doped, or heterostructured TiO_2_ material, or even other alternative catalysts [6,7,8]. Moreover, the system is compatible with any type of membrane monoliths, regardless of their characteristic pore size and the number of internal channels. However, the most significant advantage of the PNFR is the sprawling of titania via strong attachment on the entire external surface (both shell surface and lumen surface inside the channels) of the monolithic membrane [9]. In such a way, the catalyst prevents the membrane’s fouling and at the same time it cannot be released to the environment [10].

Until now, the PNFR has been tested against common azo-dyes pollutants, but the titania photocatalysts are very effective at the removal of a plethora of contaminants, such as pharmaceuticals, pesticides, or other municipal and industrial pollutants, suggesting its potential use as an innovative wastewater purification technology. Focusing on pesticides as target pollutants, the neonicotinoid insecticide acetamiprid (ACT) and the azole fungicide thiabendazole (TBZ) are very common substances, frequently used on Spanish and Greek fruit crops. Even if ACT and TBZ are allowed from the EU legislations without restrictions [11,12,13], many studies claim that they are suspicious for toxicity effects on mammals, bacteria, or fish after chronic exposure or in combination with other pesticides [14,15,16,17]. For this reason, the efficient agro-industrial wastewater treatment, the complete elimination of these pesticides and the safe disposal of the effluents is of great importance, in order to avoid any environmental threat.

In this study, a novel pilot-scale photocatalytic nanofiltration reactor was designed and fabricated for the demands of agricultural wastewater purification and reuse [18]. The PNFR pilot unit consisted of twelve seven-channeled photocatalytic nanofiltration monoliths, prepared via a wash-coating technique. The smart design of the reactor’s internals endows the process with additional flow channels, which, in addition to the monoliths, hosted 240 polyvinylidene fluoride (PVDF) porous hollow fibers with embedded TiO_2_ photocatalyst nanoparticles into their matrix. The PNFR unit enclosed 1.5 m^2^ of three photocatalytically active surfaces, achieving their irradiation on both the shell side of the monoliths and hollow fibers and on the lumen side of the monoliths with a dual illumination system. This system consisted of UV-emitting sleeved lamps, and a high-power UV LED chip module coupled with a bundle of side-glowing optical fibers. Independent photodegradation batch experiments proved the selectivity of the different photocatalytic surfaces; the removal of common organic molecules was preferable onto the hollow fibers, while the elimination of recalcitrant substances was achieved by the monoliths’ photocatalysts. This property highlights an additional asset of the PNFR unit design related to the capacity of bespoke controlling of the photocatalytic activity in the different sections of the module, to endow it with specificity for a class of compounds existing in a complex water matrix. In the case of this study, the achieved specificity was a major advantage for the efficient and long-term operation of this reactor system, because the PVDF/TiO_2_ hollow fibers acted as a wastewater pre-treatment step and the organic load reaching the monoliths was attenuated. Finally, the PNFR had a clean water production capacity of 1.2 m^3^/day, and demonstrated ~41.5% removal of TBZ, a recalcitrant antifungal agent an antiparasiticide, on the permeate effluent, without any retentate condensation, after 2 h of operation. These preliminary evaluation results shown that a 95% water recovery could be achieved under consecutive recycling and treatment steps of the effluent.

## 2. Materials and Methods

The design and construction of the pilot-scale PNFR was based on the solid grounds of a multitude of studies and knowledge acquired through massive experimentation with a lab-scale reactor prototype, described in detail previously [9,19]. In this context, a process engineering and optimization study of a scaled-up prototype resulted in the detailed design and construction of the pilot PNFR unit [18]. Briefly, this reactor consisted of ceramic tubular multi-channeled photocatalytic membrane monoliths, surrounded by porous polymeric hollow fibers (HFs) with embedded nanoparticles (photocatalyst-titania), and was equipped with appropriate flow and pressure control and illumination systems. After the conduction of process simulation studies and the assumption that the system would treat efficiently 0.6–1.0 m^3^/day of wastewater [18], the optimum number of ceramic membrane monoliths was estimated as equal to 12 (effective length 40 cm, outer diameter 2.5 cm), with 7 cylindrical channels of 6 mm internal diameter. The exact design of the reactor module, the addition to the photoactive materials, and the electronic connections with the necessary equipment will be disclosed in a detailed description on an imminent PNFR patent. In this work, only the modification of the ceramic membranes and the synthesis of the polymeric membranes will be discussed.

The transformation of the membrane monoliths to photocatalytic membrane monoliths was achieved by employing a simple and scalable wash-coating technique, which was based on a previously described method, adapted to our needs, and optimized via slight modifications [20]. In brief, 33.9 mmol of titanium (IV) isopropoxide (TTIP) were slowly added to 0.35 L of double-distilled water, at 40 °C. Then, concentrated HNO_3_ (0.05 M) was added dropwise under vigorous stirring at 80 °C, in order to catalyze the TTIP hydrolysis and obtain a transparent TiO_2_ colloidal solution after 16 h. Hence, the overall reaction, including the hydrolysis and condensations steps of the titanium precursor (TTIP) for TiO_2_ production, is summarized by the following equation:Ti{OCH(CH_3_)_2_}_4_ → TiO_2_ + 4CH_3_CH_2_CH_2_OH(1)

Once the sol was cooled down to room temperature, the commercial titania photocatalyst Evonik P25 Aeroxide (24.2 g) was gradually added, and the resulting suspension was stirred overnight until homogenization. Finally, the membranes’ modification was performed via the monolith immersion into the photocatalyst’s slurry for 10 min and a subsequent annealing step at 150 °C overnight, employing custom-made lab furnaces. In addition, two different nanoparticulate powder catalysts, named as sol and sol/P25, were obtained by drying the precursor solution and suspension at 150 °C, respectively (Figure 1).

An important asset of the PNFR reactor design was the incorporation of 120 m of porous PVDF HFs with embedded TiO_2_ nanoparticles. The photocatalytic HFs (PHFs) were prepared by the dry-jet wet phase inversion process in a spinning set-up, which has been described in a previous work [21]. Based on Dzinun et al.’s works [22,23], preliminary experiments were performed and the optimum experimental parameters of spinning PHFs were defined.

In order to get good dispersion and homogeneous distribution of the photocatalyst TiO_2_ in the dope (polymeric) solution, 3 g of TiO_2_ were dissolved gradually in 82 g of DMAc (solvent) under bath sonication for 40 min. Similarly, after the drying of the PVDF powder overnight at 120 °C under vacuum, 15 g were added progressively into the resulting solution and the suspension was stirred vigorously under mechanical stirring for at least 24 h until complete homogenization. The resulting PVDF/TiO_2_ solution was then left for 30 min in an ultrasonics bath and, before spinning, it was allowed to degas inside a stainless-steel vessel for 6 h. The bore fluid was 100% ultrapure water (Milli-Q, 18 MΩ·cm). The dope solution and the bore liquid were simultaneously pumped with a controlled flow rate (3.3 and 1.5 mL/min, respectively) through a stainless-steel tube-in-orifice spinneret employing high-precision, high-pressure gear pumps. The dope solution was filtered during the spinning procedure through a custom-made filter system, consisting of two 0.5 mm and one 0.15 mm stainless-steel sieves to remove impurities and aggregates. The internal and external diameter of the spinneret (needle) was 0.5 mm and 0.7 mm, respectively, and the orifice had a 1.2 mm internal diameter. The freshly extruded PHFs entered with an air gap (10 cm) into the coagulation bath (filtered tap water) at room temperature. The nascent PHFs were oriented by means of two guiding wheels and pulled by a cylindrical drum into a collecting tank. In order for gelation to complete and to remove any residual solvent, the produced PHFs were stored in the collecting tank containing filtered tap water for 24 h. Then, the membranes were post-treated using ethanol/water (50/50% *v*/*v*) for 1 h followed by pure ethanol for another 1 h to improve the membrane wettability and prevent pore collapse. Finally, they were hung and left to dry under tension at room temperature for 24 h.

A contact angle meter (CAM 100, KSV Instruments Ltd., Helsinki, Finland) was used for the static contact angle measurements, evaluating the surface hydrophilicity of the HFs. Deionized water (5 μL) was used for the sessile drop type and the affinity of the drop with the surface was measured applying the software fitting method. Five different specimens were employed for each sample and ten measurements were collected from different spots at room temperature (20 °C), calculating the mean contact angle and standard deviation of each sample.

The structural characterization of the prepared photocatalytic materials was performed through X-ray diffraction analysis, using a Siemens D-500 diffractometer (Siemens, Erlangen, Germany, Cu K_a1_; λ = 1.5406 Å, and K_a2_; λ = 1.5444 Å radiations). Moreover, vibrational Raman spectroscopy measurements were taken, using a Renishaw inVia Reflex spectrometer (Renishaw, Wotton-under-Edge, UK) equipped with a solid-state laser (emitting at 514.4 nm) and a Leica DMLC microscope with ×50 lens. The materials morphology was examined with a FEI Quanta-Inspect scanning electron microscope (FEI Company, Eindhoven, The Netherlands) with a tungsten filament operating at 25 kV, coupled with an energy-dispersive X-ray spectrometer (EDS). The textural and pore structural properties of the nanoparticulate photocatalysts, as well as of the developed PHFs, were evaluated by N_2_ adsorption-desorption isotherms at 77 K, employing an automated volumetric system (AUTOSORB-1 Quantachrome Instruments, Boynton Beach, FL, USA). Prior to the measurements, the samples were outgassed at 180 °C (120 °C for PHF sample) for 48 h under high vacuum achieved via a turbomolecular pump.

The specific surface area (S_BET_) was calculated by the Brunauer–Emmett–Teller (BET) method, while the pore size distribution was determined by the Barrett–Joyner–Halenda (BJH) method based on a modified Kelvin equation and the non-local density functional theory (NLDFT) methods. The average pore size (nm) was calculated from the porosimetry results as 4000·TPV/S_BET_ [24] and the mean particle size (nm) as 6000/S_BET_·d_sample_ [24], where TPV is the total pore volume and d_sample_ is the density of the sample, assuming cylindrical pore geometry of the empty space between the packed, spherical TiO_2_ nanoparticles.

The photocatalytic performance of the synthesized sol and sol/P25 samples, as well as the reference sample (P25) were evaluated via the photodegradation of the organic azo-dye methyl orange (MO, 20 ppm), a typical anionic pollutant used to evaluate the efficiency of photocatalysts, whereas for the assessment of the photocatalytic performance of the developed PHFs, rhodamine B, a typical cationic dye (RhB, 5 ppm), was employed. The experiments were performed inside a black box photoreactor, under the UV-A illumination by four Sylvania lamps (15W/BLB, 350–390 nm, ~0.5 mW cm^–2^), with a photocatalyst dosage equal to 0.1 g/L into 10 and 30 mL pollutants’ solutions for the powder and PHFs cases, respectively. At this point, it could be mentioned that the calculated dosage of the PHF sample was based on the percentage of TiO_2_ (photocatalyst) weight fraction in the developed PHFs and not on the entire mass of the sample. The photocatalytic kinetics were monitored by measuring the characteristic absorption peak of MO and RhB at 464 and 553 nm, respectively, with a Hitachi 3010 UV-Vis spectrophotometer (Hitachi Ltd., Tokyo, Japan), equipped with an integrating sphere and BaSO_4_ as reference, every 15 min for a total period of 90 min. Prior to the illumination, the solutions had been kept in the dark for 30 min, in order to achieve the dyes adsorption–desorption equilibrium onto the photocatalyst.

For the evaluation of the PNFR system, two pesticides, (acetamiprid) ACT and (thiabendazole) TBZ, which are frequently detected in the wastewater of the fruit industry, as model pollutants were employed at various concentrations (6–55 ppb) into 70–105 L initial solutions (feed tank) under different feed flow rates (2.3–3.3 L/min) and transmembrane pressures (TMP) (3.0–4.5 bar). In Figure 1 the water flow diagram of the PNFR process is depicted. The PNFR unit operated in a tangential flow mode, where the wastewater was conveyed from the feed tank to the reactor. After being subjected to all of the treatment stages (photocatalysis/concurrent photocatalysis and nanofiltration/photocatalysis) by crossing the reactor module first upwards and then downwards, driven and distributed by the specially designed reactor’s internals, the wastewater exited from the retentate outlet and returned into the feed tank, while the filtrate was drained from the bottom of the reactor and collected in the permeate tank. As such, a fraction of the volume of water that had not passed through the pores of the membrane (the concentrate) was continuously recycled, until the recovery of about 95% of the water volume that was initially in the feed tank. Calculating the residence time of an infinitesimally thin slice of the water column of height dh, it was possible to define the number of reactor’s crosses (number of cycles) until the end of the experiment. Moreover, it should be noted that since the retentate effluent had been subjected to at least two stages of photocatalytic treatment until its exit from the module, it came out less concentrated in the pollutants, an attribute that constitutes a major advantage of the PNFR process as compared to the conventional nanofiltration of water with membranes.

In order to determine the micropollutants’ decomposition rate, the samples from the photocatalytic process were analyzed by UHPLC-QqLIT-MS/MS (Agilent Technologies, Foster City, CA, USA). The chromatographic procedure is fully described in the Appendix A. Finally, the photodegradation efficiency (%) was calculated from:ΔC_0_ (%) = (C_0_ − C)/C_0_ × 100%(2)
where C_0_ is the pollutant concentration after adsorption equilibrium (mg/L) and C represents the concentration at any time during the experiment (mg/L).

## 3. Results

### 3.1. Physicochemical Properties of the PNFR’s Photoactive Components

#### 3.1.1. Nanoparticulate Titania Photocatalysts

The degree of crystallinity and the type of crystal phases of the sol and sol/P25 samples were elaborated by XRD analysis. The respective XRD diffractograms are presented in Figure 2a; both materials were mainly crystallized in the anatase TiO_2_ polymorph (JCPDS Card No.21-1272). In the case of the sol-derived sample, there existed wide and blue-shifted anatase diffraction peaks, along with the co-existence of the meta-stable brookite phase reflections (JCPDS Card No.15-0875), features inferring the inability of the employed synthesis route, due to low annealing temperature, to further promote the crystallization process. Instead, the sol/P25 presented intense diffractions of the anatase (101) plane at approximately 25° and rutile (110) plane at 27.3° (JCPDS Card No.21-1276). It is noted that both crystal planes and diffraction peaks in Figure 2a corresponded to data extracted from the Joint Committee of Powder Diffraction Standards (JCPDS). The diffraction peaks intensities were comparable with those of P25 powder, suggesting that the crystallinity of the commercial photocatalyst was not affected during the synthesis of sol/P25. Based on the reflection intensities, the weight fractions of the anatase and rutile phases were determined and were equal to 84.7 and 15.3%, respectively [25] (Table 1).

Similarly, the mean size of anatase and rutile crystallites of the sol/P25 sample was derived from Scherrer’s formula:t = K × λ/Β × cosθ(3)
with K = 0.89 (shape factor), B = breadth of diffraction reflection peak at half the maximum intensity (FWHM) and λ = 0.154056 nm (X-ray wavelength), converge with these of the P25 sample.

In addition, the findings of the vibrational Raman spectroscopy were in agreement with XRD analysis (Figure 2b). Both samples presented the E_g_, B_1g_, and A_1g_ modes of anatase phase [26], where the main vibration band of the partially crystallized sol sample was blue-shifted and characterized by enhanced FWHM. As expected, the rutile traces of the commercial P25 photocatalyst could not be easily distinguished with Raman spectroscopy, apart from an indistinct shoulder at ~450 cm^−1^ (inset of Figure 2b) [27].

The SEM examination revealed significant variations among the synthesized sol and sol/P25 materials. Firstly, the synthesized sol powder embodied thick and smooth aggregates, with size exceeding the micro-scale (Figure 3a). Smaller grains have also appeared at random spots, while inter-aggregate spaces of 2–6 μm are also present (Figure 3d). The reference P25 powder consisted of aggregated or cluster nanoparticles being assembled in formations of irregular size with almost uniform spherical shape (Figure 3b). Despite this, high surface area and an important roughness were obvious in the SEM analysis conducted at higher magnification, unveiling that the surface layers of the sample were significantly heterogeneous and porous (Figure 3e). Interestingly, the images of the sol/P25 sample could be considered as being produced from a material formed by blending the bulk sol sample and the nanoparticulate P25 photocatalyst (Figure 3c,f). In particular, by mixing the commercial titania with the precursor solution, the sol sample has been partially covered by titania nanoparticles (P25), the grain size has been shrunk, and the surface has become notably coarser.

The pore structural and textural properties of the synthesized sol and sol/P25 materials were benchmarked against those of the commercial photocatalyst (P25) by comparing the results of Liquid N_2_ (LN_2_) porosimetry at 77 K (Figure 4a). The pore size distribution (PSD) of the samples was determined from the LN_2_ isotherms desorption branch via the BJH and NLDFT methods. For the NLDFT method, an equilibrium kernel for silica as adsorbent and N_2_ 77 K as adsorbate was employed.

Overall, the LN_2_ porosimetry analysis elucidated the pore structural and textural properties of the samples in more detail (Table 2), while confirming the SEM observations. It should be noted that LN_2_ porosimetry is applicable to pore sizes ranging from 0.35 to 500 nm. Thus, the hereinafter presented analysis related to the inter-aggregate space within large and closely structured aggregates and to the interparticle pores of TiO_2_ nanoparticles.

Hence, the cumulative adsorbed LN_2_ volume at 0.97 P/P_0_ corresponded to the pores formed due to the close-packing of spherical TiO_2_ nanoparticles having a high number of contact points; whereas the volume adsorbed at the P/P_0_ interval between 0.97 and 0.99, was associated with the empty space between larger TiO_2_ nanoparticles aggregates (macropores) [24].

The sol/P25 sample presented a typical type IV isotherm, with a H1-type hysteresis loop, implying either a material with well-defined cylindrical pores or a material consisting of agglomerates of approximately uniform spheres [28], which is the characteristic structure of most mesoporous materials. The hysteresis loop observed at the end (at P/P_0_ > 0.9) was related to the capillary condensation of nitrogen in mesopores. On the contrary, the sol sample exhibited a type IV isotherm with a slight H4-type hysteresis loop, which is characteristic of pore sizes located in the boundary between micropores and mesopores, whereas the steep drop of the desorption branch at about 0.5 (P/P_0_) referred to the higher population of a specific size of pores (about 20 Å), as depicted in Figure 4a. On the other hand, the reference sample (P25) gave a type II isotherm, with a sharp capillary condensation step at a high relative pressure of P/P_0_ > 0.97, which, according to the Kelvin equation, is characteristic of materials containing large mesopores and macropores. These results demonstrated that the sol nanoparticles have partially covered the spherical P25 nanoparticles in the case of the sol/P25 material. The PSD curves (Figure 4b) of the sol/P25 sample denoted a complex pore structure with the existence of a broad range of pore sizes, while the sol sample presented narrower pore size distribution. In addition to the mesopores and macropores, the sol-derived sample also possessed smaller mesopores, an attribute that also characterized the reference P25 sample, while in the sol/P25 sample, the smaller pores were drastically reduced and larger ones were formed in the inter-aggregate space.

The degradation kinetics of MO (20 ppm) under UV-A irradiation for 90 min (Figure 5) showed that ~30% of the pollutant have been removed in the presence of the sol/P25 photocatalyst, while the sol sample presented poor photo-activity against the azo dye. It was pointed out that MO presented negligible adsorption onto the photocatalysts’ surface, so the removal effects were exclusively due to photocatalytic processes. Compared to the reference P25, the sol/P25 efficiency was slightly lower, possibly because the co-existing sol nanoparticles retarded the efficient charge carriers’ separation, thus limiting to a small extent the formation of reactive radicals. The solid–solid interface [29], the plausible heterojunction structure [30], or the interwoven anatase-rutile nanocomposite structure, as well as the synergistic effect of the co-presence of anatase and rutile endorsing the electron transfer and stabilizing charge separation, render great photocatalytic activity on P25 [31,32,33] and sol/P25 samples. In addition, for these samples, it could be deemed that the mass ratio of rutile to anatase phase similarly influenced the photocatalytic activity enhancing the transfer and separation of the photogenerated electrons and holes.

As regards the sol sample, the low crystallinity and the existence of crystal defects, despite being the cause of a large surface area, concluded to moderate photoactivity. Fluorescence spectroscopy measurements were also performed and the obtained data were summarized in Appendix A with detailed analysis and discussion in the Appendix A. The results showed that the sol sample presented a relatively lower photoluminescence (PL) intensity with respect to that of P25 and sol/P25 materials. This is probably associated with its main anatase crystalline phase, which might lead to differences in the optoelectronic properties, justifying the moderate activity of this photocatalyst [34]. Consequently, it was proved that, though the large surface area is amongst the desired properties of a photocatalytic material, it is not the decisive one for enhancing its effectiveness. Furthermore, other attributes that justify the poor photoactivity of the sol sample were its lower TPV and porosity, compared to P25 and sol/P25 samples (Table 2).

Overall, it could be mentioned that the integration of P25 with the precursor titania solution is vital for the P25 adhesion onto the PNFR membrane monoliths. Hence, the photocatalytic performance of the synthesized photocatalyst, despite being a bit lower than that of P25, was considered quite satisfactory. Herein, the structural characterization and the physicochemical properties of sol/P25 established its superiority as the material of choice for the photo-activation of the monoliths and the long-term stability and performance of the PNFR.

#### 3.1.2. Asymmetric PVDF/TiO_2_ Porous Hollow Fibers (PHFs)

Cross-section images of the PHF membrane (Figure 6a) depicted a material with typical asymmetric morphology, consisting of two thick layers that were structured around finger-like voids (pores). These layers were situated underneath and acted as supports of the outer (shell side) and inner (lumen side), ultra-thin, skin layers of the PHF membrane. A fifth structural layer, enclosing sponge-like pores, nested between the layers with the finger-like pores. The PHF exhibited an outer diameter of approximately 600 μm, while its shell and lumen surfaces were extremely smooth. The wall thickness of the PHF ranged from 120 to 145 μm, whereas the texture of the cross-section surface was quite uniform along the entire thickness and was mainly characterized by the presence of uniformly distributed titania nanoparticles.

As observed from the SEM images obtained at a higher magnification (Figure 6b,c), the outer layer presented densely arranged finger-like voids with longer and wider dimensions (about 60 and 9.5 μm, respectively) compared to those of the finger-like voids located in the inner layer (approximately 14 and 2 μm, respectively). It should be noticed that the finger-like structure could be attributed to the strength of the non-solvent (water) involved both as bore fluid and in the coagulation bath, which, due to being highly strong, was aggressively exchanged with the solvent used in the dope solution (DMAc) [35]. Furthermore, in Figure 6d, the entire sponge-like pore structure disposing big macropores with sizes ranging from 1 μm to 180 nm was presented, and seemed to be characterized by a dense and fairly uniform distribution of titania nanoparticles with few spots of aggregates.

Finally, the embodiment of TiO_2_ nanoparticles into the membrane matrix has smoothened the surface roughness (Figure 6e,f) and enhanced the membrane’s hydrophilicity. The latter was verified by measuring the water contact angle, which was reduced by 14% as compared to the neat PVDF HFs. Indeed, the water contact angle was decreased from 95.3° to 82.2° (PHF sample).

EDX analysis was performed in conjunction with SEM to identify the elemental composition at selected regions on the cross-sectional surface of the PHF (Table 3). Firstly, the detection of Ti in the prepared PHFs evidenced the effective embodiment of titania nanoparticles into the PHF matrix. Using the results from at least five different target areas of each region (near the shell, intermediate, near the lumen), the average titania loading on the cross-sectional surface was calculated at 15.5%, ranging between 15.3% and 17%. Hence, the output of EDX analysis was in very good agreement with the expected percentage (16.67%), as derived from the weight ratio of the involved precursors and the real weight of the developed PHFs. Furthermore, the titania loading was much lower in selected cross-sectional surface regions on the shell and lumen layers of the PHF. The cause of this asset was the higher affinity of titania for water as compared to the polymer. As such, during the solvent exchange process, titania nanoparticles were transferred to the water phase, but their diffusion towards the lumen and shell surfaces was significantly hindered by the aggressive (transient) invasion of water.

Regarding the Raman analysis of the neat polymer, typical bands of PVDF were illustrated in Figure 7a. In specific, the band at 458.4 cm^−1^ belongs to A_g_ symmetry corresponding to the α-phase of PVDF [36]. The band at 841.3 cm^−1^ was induced by the out-of-phase combination of CH_2_ rocking and CF_2_ stretching mode [37]. This mode is mainly attributed to the β-phase or form I of PVDF and is typical for the all-trans conformation of the PVDF chains [37]. The band at 1433.5 cm^−1^ was derived from the bending CH_2_ vibrations [38], presented mainly in the β- and γ-phases of PVDF. In addition, the band at 2979.4 cm^−1^ with high intensity is usually attributed to CH_2_ symmetric stretching [38], commonly found in the β-phase of PVDF. Finally, the band at 3019.3 cm^−1^ with strong intensity is attributed to A_g_ mode induced by the CH_2_ antisymmetric stretching [39]. As such, the β-phase of PVDF dominated on the entire HF structure, something that is consistent with recent studies on the preparation of PVDF membranes and is usually attributed to the use of pure water as the coagulation medium [40,41].

On the other hand, the anatase Raman bands were observed at 145.4 (E_g_), 400.1 (B_1g_), 518 (superposition of A_1g_ and B_1g_ modes), and 641.1 (E_g_) cm^−1^ for the PVDF/TiO_2_ sample, evidencing the successful incorporation of titania nanoparticles into the polymer matrix. The PVDF bands were also present, though with lower intensity, being overlapped by the uniform distribution of titania.

Furthermore, the PVDF/TiO_2_ HF sample (Figure 7b) gave a typical type II N_2_ adsorption isotherm (77 K), which was similar to that of the P25 sample (Figure 4a), presenting a sharp capillary condensation step at high relative pressures of P/P_0_ > 0.97, characteristic for materials containing large mesopores and macropores. In addition, the hysteresis loop at the desorption branch indicated the existence of mesoporous structure, as elucidated from the PSD in Figure 7b (inset). Indeed, a complex pore structure was revealed with the existence of a broad range of pore sizes, mainly large mesopores and macropores. It was also remarkable that while the bare P25 sample possessed pores with a size less than 9 nm, in the PVDF/TiO_2_ HF sample these pores were eliminated. Overall, the titania nanoparticles in PHFs mostly retained the structure of the parent material (P25) holding 80% of the total pore volume of P25, as observed in Table 2. It was also noteworthy that the ratio of the mean particle size D_particle_ over the mean pore size d_mean_ was higher than three (Table 2), characteristic for the random packing of equally sized hard nanospheres produced by cold isostatic pressurization [42]. This result validated the achievement of uniform distribution of the titania nanoparticles over the polymer matrix.

Finally, concerning the photocatalytic efficiency evaluation of the PHFs, a batch experiment was employed (triplicate) in order to investigate the degradation efficiency for RhB in a solution of 5 ppm. The amount of PHFs added in the testing RhB solutions was as much as required to have a titania concentration of 0.1 g/L. Hence, the experiments were performed under similar conditions with those applied for testing the powder samples (Section 3.1.1).

As shown in Figure 8a, the developed PHFs presented a moderate adsorption capacity for the dye (2.2 mg/g), determined from the attenuation of the main RhB (cationic dye) absorbance peak (553 nm). Subsequently, under UV-A illumination (Figure 8b) the immobilized photocatalyst reached a RhB removal of about 15% from the water solution at a short contact time (90 min). This performance was identical to the photocatalytic performance reported in the recent literature for the system P25-TiO_2_/RhB [43,44]. In addition, upon completion of every cycle, reweighing of the PHFs after drying provided an indication of the resistance to attrition and leaching of photocatalyst under conditions of vigorous stirring. The almost zero weight loss of the PHFs indicated very good mechanical properties of the prepared material. On the other hand, the photocatalytic properties were also investigated, after regeneration of the PHFs (reclaiming, rinsing twice with ultrapure water, and drying). Five photocatalysis/regeneration cycles were performed and the results were presented in Appendix A in the Appendix A. Overall, the PHFs preserved their activity and only a slight decrease (about 5%) of the photocatalytic performance was observed. Thus, apart from maintaining the photocatalytic efficiency of the embedded photocatalyst, the photocatalytic PHFs presented a multitude of other advantages that render them ideal candidates for their incorporation into the hybrid PNFR unit working synergistically with the nanofiltration membrane monoliths. These include their facile upscaling via the proper elongated dimension disposing a high surface-area-to-volume ratio, high flux, and very high packing density features, as well as their specificity towards pollutants abatement.

### 3.2. Design and Construction of PNFR Unit

The external appearance of the PNFR unit shares common characteristics with a typical nanofiltration (NF) module (a pressure vessel that hosts a multitude of ceramic multi-channeled monoliths), which receives a wastewater stream as feed and has retentate and permeate effluents as output, while operating in a tangential flow mode. However, the core of the PNFR module was a really complex system that included three distinct flow channels and three different active photocatalytic surfaces; the shell surface of the monolith, the lumen surface of the channels, hollow fibers and a multitude of side-glowing optical fibers, and UV sources. The particular design and construction of the PNFR unit could be divided into five different phases: (i) the fabrication of the PNFR stainless-steel, plastic, and glass components (housing, flanges, fittings, stencils, sleeves); (ii) the wash-coating technique for the photocatalytic modification of the twelve tubular ceramic membrane monoliths using the precursor suspensions of TiO_2_ photocatalyst; (iii) the development of elongated PVDF HFs, with embedded TiO_2_ photocatalyst; (iv) the assembly and sealing of the metallic, plastic, and glass components with the in-parallel settlement of the photocatalytic membranes inside the metallic housing, surrounded by the PHFs of equal length; and (v) the electrical connection of the required illumination equipment/system and monitoring sensors. Among these stages, the synthesis and the photodegradation activity of the PVDF/TiO_2_ hollow fibers have already been discussed in the previous sections. Thus, only the photo-activation of the membrane monoliths was examined in depth and evaluated as a potential and reproducible technique for the preparation of photocatalytic surfaces.

The aforesaid results of sol/P25 particulate samples have already demonstrated that the precursor P25 suspensions provide efficient photocatalysts. Thus, it was expected that the wash-coating method with this suspension would modify the membrane monoliths appropriately, by creating titania layers onto the shell surface of the monoliths and the lumen surface of the monoliths’ channels. The exact reagents ratio was slightly adapted until sufficient homogeneity of the coatings could be achieved, and then their characterization followed. The large size of the monoliths limited the structural characterization only by Raman spectroscopy and the examination was focused in three distinctive regions; the shell surface of the monolith, the cross-section area of the walls, and the lumen surface of the channels (Figure 9a). The respective areas were also examined with SEM microscopy.

As shown in Figure 9b, the monolith surface and the cross-section areas of an uncoated membrane demonstrated the typical 2A_1g_ + 5E_g_ modes of a-Al_2_O_3_ (corundum), with predominant A_1g_ vibration at ~417 cm^−1^ [45,46]. In general, the channels surface was pre-coated with an NF thin layer by the manufacturer, possibly consisting of ZrO_2_. However, no vibration was detected, implying the amorphous nature of the NF layer. On the other hand, the examination of the titania-modified membrane revealed extra bands, arising from the anatase TiO_2_ polymorph (Figure 9c). The detection of the main E_1g_ band at ~143 cm^−1^ on both the shell and lumen surfaces proves the successful incorporation of the titania photocatalyst onto the ceramic membrane. Contrary to our previous study [9], the catalyst did not penetrate in depth inside the bulk part of the membranes body, possibly due to the much higher viscosity of the wash-coating slurry used in this work, as compared to the titania precursor sol of the previous development. However, extending the titania deposition further in depth was not expected to have any essential effect on the photocatalytic performance of the PNFR unit due to light shielding effects.

The investigation of the membranes with SEM microscopy showed that the coverage with titania was uniform across the membranes’ surfaces (Figure 10). Before the wash-coating, the shell surface of the monoliths seemed like a random arrangement of porous Al_2_O_3_ plates, while the NF layer on the lumen surface of the channels was smooth and compact (Figure 10a–c). Upon the addition of TiO_2_, titania superficially covered the alumina plates and the NF layer and filled all the empty space on the shell surface of the monolith (Figure 10d,f), whereas changes on the bulk part were not obvious (Figure 10e).

By determining the elemental weight percentage using the EDS technique (Table 4), it was revealed that the surface coverage was sufficient in the upper edge and the middle part of the membrane, while the deposition on the bottom edge was favored significantly during the wash-coating process. The respective SEM images of the channels depicted the deposition of a thick, smooth, and homogenous TiO_2_ coating onto the NF layer (Figure 10e), with a uniform distribution along the membrane (Table 4). Lastly, the cross-section examination of both reference and unmodified membranes were corroborating the Raman analysis; no titania was detected in the bulk part of the membranes, either with the SEM or the EDX analysis.

### 3.3. Operation and Photocatalytic Performance of PNFR Unit

For the evaluation of the photocatalytic performance of the PNFR unit, two pesticides, ACT and TBZ, were employed as model pollutants. The specific conditions involved in our experimental campaign with the pilot PNFR unit were included in Table 5.

Specifically, the first two runs of the PNFR unit were carried out with a feed flow of 3.3 L/min and TMP of 4 bar, while the ACT concentration was maintained at 50 ppb. These two runs were conducted sequentially and without the application of a cleaning procedure in between them. This was performed intentionally with the purpose to draw conclusions on the impact of the cleaning process on the pesticides removal efficiency and verify that by simply flushing the reactor with fresh water under UV illumination, it was possible to fully restore the performance. Furthermore, striving to examine the influence of the feed flow rate and TMP, which were factors effectuating radical changes on the residence time and the number of cycles of the water phase inside the reactor, the subsequent two runs were conducted with 2.5 L/min feed flow and 3 bar pressure. The contribution of adsorption was elaborated by performing a fourth run under dark conditions. For the last two runs TBZ was employed as a contaminant with the target to simultaneously investigate the adsorption and photocatalytic effects on a compound of different physicochemical characteristics.

After each experimental run with the ACT or TBZ contaminated water, the entire PNFR unit was cleaned up for 2 h with fresh tap water under UV illumination. The cleaning procedure was repeated until the complete desorption of the micropollutant, which was concluded by the achievement of positive rejection performance. Instead, when the regeneration was not complete, the C/C_0_ values were higher than 1 at both the retentate and permeate effluents, as shown in the respective plot (Appendix A) included in the Appendix A.

As depicted in Figure 11, with the attenuation of the TMP from 4 to 3 bar and the decrease of the feed flow rate from 3.3 to 2.5 L/min, the PNFR unit achieved significantly higher performance (Figure 11a,c). The ACT removal efficiency at the permeate side was amplified from 20% to about 30%, while the respective amplification at the retentate side was from 15% to 25% (Figure 11c). At this point, it should be mentioned that the lower feed flow rate enhanced the contact time of the micropollutant with the photocatalytically active surfaces nesting into the flow channels that were integrated in series from the input to the retentate output of the module. It was calculated that the reduction of the flow rate from 3.3 to 2.5 L/min induced a 3.5 min increase in the cycle time of water inside the reactor. On the other hand, the lower pressure enhanced the contact time of the micropollutant with the photocatalytic layers that were deposited on the lumen surface of the monoliths. This was a result of the lower permeation flux, which, with the drop of pressure from 4 to 3 bar, was reduced from 0.39 to 0.13 L/min. Since the reactor was assembled vertically, as soon as the water phase passed through the pores of the monoliths, it was conformed as a thin film that slips down onto the photocatalytic surface deposited on the lumen side of the monoliths. The slip down velocity correlated linearly with the permeate flow and inversely with the contact time between the pollutant and the photocatalytic surface.

In addition, the cleaning intervals of the PNFR unit had a great impact and defined to a high extent the performances that could be achieved. This was clearly shown in Figure 11b, which depicts the removal efficiencies in the permeate and retentate side during an experiment that was conducted without applying a cleaning cycle before its commencement. It can be seen that the ACT removal efficiency in the permeate and retentate effluents was almost half of those achieved in the foregoing experiment (Figure 11a).

The contribution of the ACT adsorption on the achieved photocatalytic performances was elaborated in the fourth run (Figure 11d), where the light sources were kept off. Compared with the experiment performed under UV illumination (Figure 11c), the ACT removal efficiency was 15% at the permeate side (instead of 20% with UV) and 12% at the retentate side (instead of 15% with UV). The small differences manifested the important contribution of ACT adsorption on the PNFR unit’s performance. This is a well-known feature of heterogeneous photocatalysis on solid semiconductors since the reactant species firstly need to be adsorbed on the photocatalyst’s surface. In most of the cases, photocatalysis is the rate determining stage since adsorption is very fast and occurs spontaneously. Despite this, there are further stages that complete the cycle of the photocatalytic process. Hence, the intermediate products must be desorbed and diffuse into the solution and ACT molecules being sequentially adsorbed on the photocatalyst’s surface to reach a new equilibrium. The above steps proceeded circularly until the mineralization of ACT molecules. As such, we could speculate that the cause for the small difference between the photocatalytic and adsorption performances was that the intermediate products of the photocatalytic degradation of ACT were strongly adsorbed on the photocatalyst’s surface, and consequently their desorption and diffusion back into the solution constituted the rate determining step of the overall process.

On the other hand, for the case of TBZ, being a less recalcitrant compound than ACT and exhibiting less affinity for the photocatalyst’s surface, the photocatalytic removal efficiency at the permeate and retentate effluent reached performances of 46.5% and 45%, respectively. As presented in Figure 11e, these performances were achieved after twelve water recirculation cycles. In the case of TBZ, the overall performance was mainly contributed by the photocatalytic process induced by the UV illumination, as the adsorption rendered a minor effect on the total degradation process (Figure 11f).

Another important parameter that was monitored during the accomplishment of the experimental campaigns was the evolution of the water permeance. The water permeance Pe (L/m^2^·h·bar) is a property of the membrane and is derived from the following equation:(4)$$\mathrm{Pe}=\frac{F}{P\times S}$$
where F (mL/min) is the water flux through the membrane, P (bar) is the TMP, and S (m^2^) is the surface of the membrane, which, in our case, coincides with the photocatalytic shell surface of the monoliths.

In addition, the active photocatalytic surfaces of the hollow fibers and monoliths (shell surface and lumen surface in the channels) were calculated from the respective dimensions and were 0.45, 0.38, and 0.63 m^2^, respectively.

In Figure 12a, the time evolution of the water permeance relative to the initial value was presented for each experiment. The initial water permeance ranged from 2.2 to 15.6 L/m^2^·h·bar, depending on the hydrodynamic conditions (feed flow and TMP). It is remarkable that in all experiments performed at a lower TMP and feed flow (3 bar and 2.4 L/min), the permeance of the membranes was subjected to a sudden drop down to 60% of the initial value. This happened at the very early stages of each experiment (within the first 10 min of flooding), and thenceforth, steady state conditions were established and the permeance remained constant. Despite that, the lower flow rate of the feed water led to lower crossflow velocity, which may promote the fouling propensity of the membranes; it was assumed that the sudden drop of permeance was not caused by pore blocking or cake layer formation. Besides, the low levels of turbidity and conductivity in the feed water (Figure 12b,c) would not justify a fouling event of this type. On the other hand, since the photocatalytic layers were deposited on the shell and lumen surfaces of nanofiltration monoliths with a molecular weight cut-off (MWCO) of 1000 Da, the produced photocatalytic membranes may also exhibit the capacity to reject the tested solutes, though to a low extent, with the rejection mechanism being mostly induced by Donnan rather than steric effects. The weak rejection capacity of the membranes can be concluded by the fact that during the photocatalytic runs, the % removal of the pesticides in the permeate was always higher than that in the retentate (Figure 11a–e). This is contradictory to what was expected, given that the total photocatalytic surface incorporated into the flow channels from the feed to the retentate effluent (0.83 m^2^) was much higher than that existing on the lumen surface of the monoliths at the permeate side (0.63 m^2^). Hence, there must be slight rejection of the solutes, which effectuated the achieved performances at the retentate and permeate effluents. Since there was rejection capacity for the solutes, there may be also a concentration polarization process taking place, which leads to the accumulation of retained solutes in the membrane boundary at the feed side. Concentration polarization resulted in significant reduction of the permeate flux. However, its intensity was enhanced at low crossflow velocities. The crossflow velocity of the experiments performed at a feed flow rate of 2.4 L/min was very low (0.2 cm/sec) and this justified the sudden drop of the permeance.

Focusing on the plots of permeance evolution (Figure 12a), a different trend could be distinguished between the experiment performed in the dark (fourth run) and those performed under UV illumination. Specifically, after 50 min on stream, the water flux through the illuminated membranes seemed to have an uprising trend, indicating the occurrence of photo-induced hydrophilicity effects on the titania photocatalytic layers [19].

Regarding the electrical conductivity in the feed tank, the slight increase (Figure 12b) by the salts’ presence could be justified by the low rejection efficiency concerning divalent metal cations, such as Ca^2+^ and Mg^2+^. Despite that, the prospective seemed auspicious, as it was presumed that the PNFR unit is capable of removing organic pollutants and rejecting inorganic divalent metal ions provided the use of membranes with smaller pore size. In addition, the adjustment of pH under six could render the membranes positively charged, enhancing the rejection efficiency via the Donnan effects. Hence, the dynamic of the system to degrade organic contaminants and reject divalent metal ions, and especially heavy metal ones, was truthfully promising.

The increase of turbidity in Figure 11c after a period of 60 min, and more intensively during the initial runs with higher TMP and feed flow hydrodynamic conditions, could be correlated with two factors. In specific, the aforementioned calculated permeation flux was triple during the initial two runs compared to the following two ones resulting in one-third corresponding water volume treatment involvement, and in addition, the condensate of the effluent was larger. Hereafter, the particles rejection was more intense and faster and this was depicted in the turbidity diagram.

Finally, the compression of water via the working pump and the UV illumination led to the temperature increase during the process, as observed in Figure 12d. The influence of the UV illumination system on temperature was higher, as during the runs with switched on light sources, the temperature presented as a steeper slope. Thus, the sharp raise of temperature influenced negatively on the adsorptive efficiency of the active centers of the membranes (adsorption is exothermic) and eventually resulted in the reduction of the removal efficiency.

The cycle time was calculated by the following equation:(5)$$\mathrm{Cycle}\;\mathrm{time}=\frac{h\times (S_{\mathrm{up}\;}+{\;S}_{\mathrm{down}})}{F}$$
where h (cm) is the height of the reactor and S_up_ and S_down_ (cm^2^) is the net up and down surfaces from which the flow stream passes through during the process. The net amount of the micropollutant removed (m, in mg) was extracted from the mass balance equation, that is:
(6)$$m=\int_{0}^{t}C_{f}\mathrm{Ftdt}-\sum_{i=0}^{t}{\left(C_{p}\right)}_{i}{\left(V_{p}\right)}_{i}-\sum_{i=0}^{t}{\left(C_{r}\right)}_{i}{\left(V_{r}\right)}_{i}-V_{D}\text{×}C_{f}\text{×}{\bar{\left(\frac{C}{C_{0}}\right)}}_{r}$$
where C_f_ (mg/L) is the initial feed pollutant concentration, t (min) is the time elapsed from the start of the experimental run, (C_p_)_i_ and (C_r_)_i_ (mg/L) are the micropollutant concentrations in the permeate and retentate stream, respectively, (V_p_)_i_ and (V_r_)_i_ (mL) are the liquid volumes collected from the permeate and retentate effluent at the specific time interval, and V_D_ (L) is the dead volume of the reactor. Finally, the rejection percentage was derived from the fraction of the total amount of the pollutant removed to the initial amount of the micropollutant. Accordingly, the total performance characteristics of the PNFR unit were depicted in Table 6. The maximum rejections of ACT and TBZ reached approximately the values of 25 and 41.5%, respectively.

## 4. Conclusions

In this work, a novel hybrid photocatalytic nanofiltration reactor (PNFR) pilot unit was constructed and its ability verified in recycling 1.2 m^3^/day of real agro-wastewater. These promising results constitute a motivation for further validating the replicability and transferability of the PNFR technology in various real-life purification applications.

Prior to the ceramic monoliths’ wash-coating and the PNFR assembly, the unit’s active components were thoroughly examined and their complete physicochemical and structural characterization was performed. It was revealed that by mixing the commercial titania P25 photocatalyst with the TiO_2_ precursor solution, the final powder sample created smaller aggregates of lower surface area and higher roughness, while the crystallization in the anatase phase was retained. In the case of porous hollow fibers, P25 was successfully distributed and integrated in the PVDF structure, improving the materials’ physicochemical properties. Both sol/P25 and PHFs degraded efficiently the target pollutants, even at high concentrations, ensuring that the PNFR will furnish three active and efficient photocatalytic surfaces of high specific surface area to reactor volume ratio. Indeed, the facile wash-coating technique yielded modified photocatalytic membranes, which the PNFR technology exploits to combine synergistically the photocatalytic process with the filtration treatment, such that a sequential degradation action occurs within the reactor. In addition, for evaluating the photocatalytic performance of the PNFR pilot unit, experimental campaigns were performed involving different feed flow rates (2.3–3.3 L/min) and TMP (3.0–4.5 bar) and using various concentrations of thiabendazole and acetamiprid, which are frequently detected in the wastewater of the fruit industry. In about 3 h, the amount of TBZ and ACT was reduced by 41.5 and 25%, respectively. The obtained results confirm the effective incorporation of photocatalytic nanofiltration in real-life applications. Due to the facile reactor’s upscalability, increased industrial interest is expected for the developed filtration-assisted advanced oxidation technology (AOT) in the field of wastewater purification and reuse.

## Data Availability

Not applicable.

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
