# Peer review of "Novel Pilot-Scale Photocatalytic Nanofiltration Reactor for Agricultural Wastewater Treatment"

_membranes, 2023, doi:10.3390/membranes13020202_

Round 1

Reviewer 1 Report

This work describes the use of hybrid photocatalytic nanofiltration reactor for the combined synergic effect of photocatalytic degradation and size exclusion of two selected micropollutants. The surface modification of membrane monoliths with TiO2 via wash-coating deposition is described to promote an enhance in the reactor performance. The manuscript is well-written, logically constructed, and the discussion is strongly supported by obtained data. The manuscript is interesting and relevant for Membranes journal, and I recommend it for publication. I have a few comments to be improved before publication.

Line 273/275: “As expected, the rutile traces of the 273 commercial P25 photocatalyst cannot be easily distinguished with Raman spectroscopy, 274 apart from an indistinct shoulder at ~450 cm-1”. This is not evidenced by Fig 2b.

Did the authors investigate the possible catalyst leaching or evaluate the performed characterization studies after the filtration treatment cycles?

All the Y-axes of Figure 11 should be in the same range for easier comparison.

Reviewer 3 Report

In this manuscript, Theodorakopoulos et al. described “Novel pilot-scale photocatalytic nanofiltration reactor for agricultural wastewater treatment” in detail. Samples are properly characterized, and the activities are excellent. However, at this stage there are still many problems and I therefore suggest a major review for this manuscript keeping in mind the following questions.

1) The abstract is too large and not attractive, please write a precise abstract.

2)  The English language is very poor. A definite relationship must exist between the different portions of the manuscript, for example abstract and conclusion are written in past tense, experimental part is written in past, result and discussion are written in present tense, while in introduction, shortcomings are written in present tense, review literature is written in past tense. No correlation is seen in the manuscript. For example, the sentence “Smaller grains had also appeared at random spots, while inter-aggregate spaces of 2-6 μm were also present” means that you are describing some thing in past while in fact you are showing your results to the reviewers, therefore, the correct form of the sentence is “Smaller grains has also appeared at random spots, while inter-aggregate spaces of 2-6 μm are also present”. Similarly, other sentences need careful attention.

3) What is the solubility of titanium (IV) isopropoxide in water? Does it form precipitate when added to water?

4) The sentences in some places are too long and divert the attention of the readers, the respected authors are requested to use short and precise sentences.

5) What is the light intensity of the light used for irradiation?

6) Did the author perform any test for the measurement of the intermediates formed during the decomposition of the dyes?

7) The conclusion is also too large, please cut it into a precise smaller conclusion.

8) Is it necessary that adsorption equilibrium will reach after 30 min as the author mentioned in the manuscript?

9) Some very important citations are missing.

i) A. Zada, N. Ali, F. Subhan, N. Anwar, M. I. A. Shah, M. Ateeq, Z. Hussain, K. Zaman, M. Khan, Suitable energy platform significantly improves charge separation of g-C3N4 for CO2 reduction and pollutant oxidation under visible-light, Prog. Nat. Sci. Mat. Int. 29 (2019) 138-144.

ii) B. Xu, A. Zada, G. Wang, Y. Qu, Boosting the visible-light photoactivities of BiVO4 nanoplates by doping Eu and coupling CeOx nanoparticles for CO2 reduction and organic oxidation, Sust. Energy Fuels 3 (2019) 3363-3369.

Round 2

Reviewer 3 Report

Since the authors made significant changes and improved the quality of the paper by responding well to all my questions and carrying out necessary changes, I therefore, accept the publication of this paper in your reputed journal.